# Protein Panel of Serum-Derived Small Extracellular Vesicles for the Screening and Diagnosis of Epithelial Ovarian Cancer

**DOI:** 10.3390/cancers14153719

**Published:** 2022-07-30

**Authors:** Huiling Lai, Yunyun Guo, Liming Tian, Linxiang Wu, Xiaohui Li, Zunxian Yang, Shuqin Chen, Yufeng Ren, Shasha He, Weipeng He, Guofen Yang

**Affiliations:** 1Department of Gynecology, The First Affiliated Hospital, Sun Yat-Sen University, Guangzhou 510080, China; laihling@mail.sysu.edu.cn (H.L.); guoyy79@mail.sysu.edu.cn (Y.G.); tianlm3@mail2.sysu.edu.cn (L.T.); wulx3@mail2.sysu.edu.cn (L.W.); lixh36@mail2.sysu.edu.cn (X.L.); yangzunxian666@163.com (Z.Y.); 2Department of Gynecology, The Six Affiliated Hospital, Sun Yat-Sen University, Guangzhou 510080, China; chshqin@mail.sysu.edu.cn; 3Department of Radiation Oncology, The First Affiliated Hospital, Sun Yat-Sen University, Guangzhou 510080, China; renyuf@mail.sysu.edu.cn (Y.R.); heshsh23@mail.sysu.edu.cn (S.H.)

**Keywords:** ovarian cancer, extracellular vesicle, proteomics, ovarian cancer screening, differential diagnosis

## Abstract

**Simple Summary:**

Ovarian cancer is a common gynecological malignancy, with the highest fatality rate. At the time of this study, there were no available biomarkers with high sensitivity and specificity for screening ovarian cancer. Our study provided a proteomic signature of circulating small extracellular vesicles derived from the serum in ovarian cancer. The diagnostic proteomic panel constructed for ovarian cancer may complement current clinical diagnostic measures for screening ovarian cancer in the general population and the differential diagnosis of ovarian masses.

**Abstract:**

Although ovarian cancer, a gynecological malignancy, has the highest fatality rate, it still lacks highly specific biomarkers, and the differential diagnosis of ovarian masses remains difficult to determine for gynecologists. Our study aimed to obtain ovarian cancer-specific protein candidates from the circulating small extracellular vesicles (sEVs) and develop a protein panel for ovarian cancer screening and differential diagnosis of ovarian masses. In our study, sEVs derived from the serum of healthy controls and patients with cystadenoma and ovarian cancer were investigated to obtain a cancer-specific proteomic profile. In a discovery cohort, 1119 proteins were identified, and significant differences in the protein profiles of EVs were observed among groups. Then, 23 differentially expressed proteins were assessed using the parallel reaction monitoring in a validation cohort. Through univariate and multivariate logistic regression analyses, a novel model comprising three proteins (fibrinogen gamma gene (FGG), mucin 16 (MUC16), and apolipoprotein (APOA4)) was established to screen patients with ovarian cancer. This model exhibited an area under the receiver operating characteristic curve (AUC) of 0.936 (95% CI, 0.888–0.984) with 92.0% sensitivity and 82.9% specificity. Another panel comprising serum CA125, sEV-APOA4, and sEV-CD5L showed excellent performance (AUC 0.945 (95% CI, 0.890–1.000), sensitivity of 88.0%, specificity of 93.3%, and accuracy of 89.2%) to distinguish malignancy from benign ovarian masses. Altogether, our study provided a proteomic signature of circulating sEVs in ovarian cancer. The diagnostic proteomic panel may complement current clinical diagnostic measures for screening ovarian cancer in the general population and the differential diagnosis of ovarian masses.

## 1. Introduction

Ovarian cancer is one of the most common gynecological malignancies, with 313,959 new cases and 207,252 deaths worldwide in 2020 [1]. The high fatality rate accounts for 2.5% of all malignancies among females, but 5% of cancer-related deaths. More than 70% of patients are diagnosed at stage III or IV because of a lack in exhibiting typical symptoms and effective screening. For all stages of ovarian cancer, the 5-year survival rate is approximately 46%. The 5-year survival rate is less than 20% in women diagnosed with advanced stage (Federation of Gynecology and Obstetrics (FIGO) stage III or IV) invasive epithelial ovarian cancer (EOC); however, it exceeds 90% for those at stage I [2]. Therefore, increasing early ovarian cancer detection is the primary strategy to improve ovarian cancer patient survival.

Ovarian cancer is highly heterogeneous and demonstrates various molecular, pathological, and other features. EOC is the most common histomorphological type and accounts for more than 90% of ovarian cancer. Currently, transvaginal ultrasound scan (TVS), level of serum carbohydrate antigen 125 (CA125), and pelvic examination are the most commonly used EOC diagnostic methods [3]. Unfortunately, pelvic examination lacks objective data standards and depends on the proficiency of the attending physician. Moreover, patients with early disease are likely to be overlooked because of the lack of typical signs. TVS can provide valuable information on an ovarian mass, including its size, location, composition, morphology, and blood flow, and these features are frequently used to assess malignancy. Although TVS has a sensitivity of more than 90% for the differential diagnosis of ovarian masses, its performance largely depends on the skill and experience of the ultrasound specialists [4]. Furthermore, CA125 is elevated in only about 80% of EOC and is associated with other diseases, resulting in an unsatisfactory diagnostic performance when served as the sole diagnostic indicator [5]. Alternative to the single current indicator, many studies have also proposed the comprehensive model of multiple indicators for diagnosing ovarian cancer, including the risk of malignancy index (RMI), the risk of ovarian cancer algorithm (ROMA), and the One Variable-at-a-Time (OVAT1) [6,7,8]. However, the application of larger population data demonstrated that these combined models failed to produce a more stable and reliable diagnostic performance than a single index. Therefore, a new biomarker with superior sensitivity and specificity is still urgently needed for ovarian cancer.

Extracellular vesicle (EV) is a generic term for particles naturally released from cells that are delimited by a lipid bilayer and cannot replicate. Small EVs (sEVs), with a diameter of <200 nm, are present in various bodily fluids such as blood, urine, saliva, and ascitic fluid [9]. By delivering nucleic acids, proteins, lipids, ions, and other specific components, sEVs participate in different pathophysiological processes to maintain cell homeostasis and regulate intercellular communication [10]. The diverse effects mediated by tumor-derived sEVs have gradually been discovered, including epithelial–mesenchymal transition, metastasis, angiogenesis, immune regulation, cell metabolism, pre-metastasis niche formation, and therapeutic resistance [11,12]. In addition to their role as biocommunication mediators, the great potential of sEVs as circulating biomarkers for the diagnosis and prognosis of various cancers has attracted considerable attention [13]. For example, in ovarian cancer, sEVs can be isolated from the ascitic fluid and blood of patients, making them potentially less invasive biomarkers for its diagnosis and prognosis. Moreover, microRNA (miR-200a, miR-200b, miR-200c, and miR-373) levels in sEVs derived from the serum of EOC patient showed diagnostic and prognostic values in a study conducted by Meng et al. [14]. Furthermore, proteins from sEVs, including claudin-4, transforming growth factor-β1 (TGF-β1), and melanoma-associated antigen 3 and 6 (MAGE3/6), were considered as candidate biomarkers because of their enrichment in the serum of ovarian cancer patients compared to that in the patients with benign tumors or healthy controls [15,16]. However, these indicators have not yet been verified with a large sample size and developed for further clinical applications, owing to their unsatisfactory diagnostic performance or the difficulty and poor stability in detection methods.

In this study, we performed tandem mass tag-based liquid chromatography/tandem mass spectrometry (TMT-LC-MS/MS) analysis to obtain the proteomic profiles of sEVs derived from patients with EOC. Our study aimed to obtain cancer-specific protein candidates in sEVs from the serum of ovarian cancer patients and develop a protein panel for ovarian cancer screening in the general population and differential diagnosis of ovarian masses.

## 2. Materials and Methods

### 2.1. Patients and Serum Samples

In total, 66 patients with ovarian masses (16 patients with cystadenoma and 50 patients with ovarian cancer) and 29 healthy subjects from the first affiliated hospital of the Sun Yat-sen University (Guangzhou, Guangdong Province, China) were enrolled in this study conducted between January 2018 and January 2020. All human blood samples were procured after obtaining the approval of the institutional review board of the first affiliated hospital of Sun Yat-sen University and informed consent from all participants. The detailed characteristics of all the participants are provided in Appendix A. Whole-blood samples were obtained in vacutainer serum blood collection tubes (BD Biosciences, Franklin Lakes, NJ, USA) from participants who were made to fast. After coagulation, the blood samples without hemolysis were centrifuged at 3000 rpm for 10 min at 4 °C to obtain serum samples. The supernatants were aliquoted and were stored at −80 °C for subsequent sEVs’ isolation.

### 2.2. Isolation of sEVs

The serum samples were removed from storage at −80 °C and centrifuged at 12,000× *g* for 15 min at 4 °C. Next, the supernatants were transferred to a new centrifuge tube and filtered with a 0.22-μm microporous membrane. In HPLC/LC-MS/MS analysis and targeted proteomics, 1 mL aliquots of serum were subjected to the isolation of sEVs using commercially available qEVoriginal size-exclusion chromatography (SEC) columns (Izon Science, Christchurch, New Zealand), according to the manufacturer’s protocol [17]. The purified sEVs were then used immediately or stored at −80 °C.

### 2.3. Transmission Electron Microscopy (TEM)

Isolated sEVs were resuspended in 50–100 μL of 2% paraformaldehyde solution, and 5 μL of this suspension was added to the Formvar–carbon copper grip and 100 μL PBS was added to the sealing film. The copper grid was placed on PBS drops and washed. The copper grid was placed on a 50 μL 1% glutaraldehyde droplet for 5 min and then washed with 100 μL double distilled water. Next, the copper grid was placed on 50 μL uranium oxalate drops for 75 min and then placed on 50 methylcellulose drops for 10 min (placed on ice). The excess liquid was absorbed on the filter paper and the copper grid was air-dried for 5–10 min. Finally, the copper grid was placed in the box, and electron microscopy photos were acquired at 80 kV.

### 2.4. Nanoparticle Tracking Analysis (NTA)

The particle size and concentration of sEVs were measured using NTA at Viva Cell Biosciences with ZetaView PMX 110 (Particle Metrix, Munich, Germany) and the ZetaView 8.04.02 software. The isolated sEVs were diluted 500 times with PBS before measurement. NTA measurements were recorded and analyzed at 11 positions. The ZetaView system was calibrated using 110 nm polystyrene particles. Temperature was maintained between 24.41 and 25.74 °C. Each process was repeated thrice.

### 2.5. Western Blotting Analysis

Isolated sEVs from 1 mL aliquots of serum were lysed in RIPA buffer with a protease inhibitor cocktail and quantified using a BCA protein assay reagent kit (Thermo Scientific, Waltham, MA, USA). The protein samples were denatured at 95 °C for 10 min in a 5× protein loading buffer. Equal amounts of the protein extracts were separated using 10% SDS-PAGE and transferred onto a PVDF membrane (Millipore, MA, USA). The membranes were blocked with 5% evaporated skimmed milk for 1 h in TBST at room temperature and then incubated overnight with primary antibodies against the following human proteins: TSG101 (#ab125011, 1:1000 dilution, Abcam, Cambridge, UK), CD63 (#ab134045, 1:5000 dilution, Abcam, Cambridge, UK), Albumin (#GTX102419, 1:1000 dilution, GeneTex, Irvine, CA, USA), FGG (#GTX108640, 1:5000 dilution, GeneTex, Irvine, CA, USA), MUC16 (#20077-1-AP, 1:500 dilution, Proteintech, Wuhan, China), and APOA4 (#17996-1-AP, 1:8000 dilution, Proteintech, Wuhan, China) at 4 °C. The membranes were then incubated with HRP-conjugated secondary antibody for 1 h at room temperature. The protein bands were visualized using enhanced chemiluminescence detection reagents (Thermo Scientific) following the manufacturer’s instructions.

### 2.6. Protein Extraction and Digestion

For protein lysis, an 8 M urea and 1% protease inhibitor cocktail was added to the sEVs, followed by sonication thrice on ice using a high-intensity ultrasonic processor. Subsequently, the protein concentration was determined using the BCA kit, following the manufacturer’s instructions. For digestion, the protein solution was reduced with 5 mM dithiothreitol for 30 min at 56 °C and alkylated with 11 mM iodoacetamide for 15 min at room temperature in the dark. The protein sample was then diluted by adding 100 mM TEAB to achieve a urea concentration less than 2 M. Finally, trypsin was added at a 1:50 trypsin-to-protein mass ratio for the first digestion overnight and a 1:100 trypsin-to-protein mass ratio for the second 4 h digestion.

### 2.7. Tandem Mass Tag (TMT) Labeling, HPLC Fractionation, and LC-MS/MS Analysis

After trypsin digestion, the peptide was desalted using Strata X C18 SPE column (Phenomenex Inc., Torrance, CA, USA) and vacuum dried. Based on the results of peptides’ quantification, the least number of peptides, 14 μg peptides, were sampled for labeling. The peptide was reconstituted in 0.5 M TEAB and processed following the manufacturer’s protocol for the TMT kit, and 2 μg peptides were sampled from each sample and mixed as MIX samples. After thorough mixing, 14 μg peptides were sampled from the MIX samples. Following the standardized quality inspection, the labeled samples of each group were mixed and subjected to HPLC fractionation. The tryptic peptides were fractionated using high pH reverse-phase HPLC using Agilent 300Extend C18 column. The peptides were subjected to the NSI source followed by tandem mass spectrometry (MS/MS) in Q Exactive^TM^ Plus (Thermo Scientific), coupled online to the UPLC. The resulting MS/MS data were processed using the MaxQuant search engine (v.1.5.2.8) (Max-Planck Institute of Biochemistry, Munich, Germany). Tandem mass spectra were searched against the SwissProt human database concatenated with a reverse decoy database.

### 2.8. Targeted Proteomics

For the validation study, sEVs isolated from 1 mL aliquots of serum were lysed and digested, as described above. After digestion, the peptides were quantified. Then, 1.5 ug peptides from each sample were analyzed with LC-MS operated using the parallel reaction monitoring (PRM) acquisition scheme, a targeted proteomics technique based on high-resolution and high-precision mass spectrometry. Finally, the resulting MS data were processed using Skyline (v.3.6) (MacCoss Lab Software, University of Washington, Seattle, WA, USA).

### 2.9. Bioinformatic Analysis

The software used for the bioinformatics analysis is listed in Appendix A. Gene ontology (GO) annotation proteome was derived from the UniProt–GOA database. Proteins were classified into three GO annotation categories: biological process, cellular compartment, and molecular function. For each category, a two-tailed Fisher’s exact test was employed to test the enrichment of the differentially expressed proteins against all identified proteins. The GO term with an adjusted *p*-value < 0.05 was considered significant [18]. The reactome pathway database was used to identify enriched pathways using a two-tailed Fisher’s exact test for differentially expressed protein. The pathway with an adjusted *p*-value < 0.05 was considered significant. Finally, the STRING database was used for protein–protein interaction (PPI) network analysis. The medium confidence was set as 0.4, and line thickness indicated the strength of data support.

### 2.10. Statistical Analyses

All statistical analyses were performed using the R version 4.1.3 (available online: https://www.R-project.org/ (accessed on 10 March 2022)) and SPSS 25.0 (SPSS Inc., Chicago, IL, USA). Continuous variables are described in terms of medians (interquartile range (IQR)). Group differences in protein expression were analyzed using the two-tailed *t*-test. ROC curve analyses were employed to assess the diagnostic efficacy. Univariate and multivariate logistic regression analyses were used to develop a diagnostic model. Estimates were presented as odds ratios (ORs) and 95% confidence intervals (CIs). All significance tests were two-tailed and conducted at a *p*-value < 0.05.

## 3. Results

### 3.1. General Experimental Design and Clinical Synopsis

A method based on proteomic analysis was designed to obtain a comprehensive understanding of the proteins present in the sEVs derived from the serum of patients with ovarian cancer, as shown in Figure 1. In the discovery study, nine patients with ovarian cancer (Ca group), nine patients with cystadenoma (a representative of benign ovarian tumor, Cys group), and nine healthy controls (HC group) were enrolled (Appendix A). In addition, 50 patients with ovarian cancer, 15 patients with cystadenoma, and 20 healthy controls were enrolled in the validation cohort. Of the 50 ovarian cancer patients, the pathological stages were distributed as follows: stage I in 12 patients, stage II in 2 patients, stage III in 25 patients, and stage IV in 11 patients (Appendix A). Participants in the discovery cohort were included in the validation cohort, except for one patient with cystadenoma, because the serum sample was not sufficient for subsequent analyses. Thus, in the validation study, the Cys group included six patients with serous cystadenoma and ten patients with mucinous cystadenoma; the Ca group included 30 patients with serous carcinoma, two patients with mucinous carcinoma, four patients with endometrioid carcinoma, nine patients with clear cell carcinomas, and five EOCs with undefined pathological type.

### 3.2. Isolation and Identification of sEVs

SEC isolated sEVs from the serum samples of healthy controls and patients with ovarian cancer or cystadenoma. Then, sEVs were characterized using NTA, TEM, and immunoblotting. TEM analysis displayed round or ellipsoidal shaped, lipid layer-enclosed vesicles ranging 80–200 nm in diameter (Figure 2A). NTA revealed that the average size of the purified sEVs was 176.4 ± 2.9 nm, and the primary peak size was at 120 nm (Figure 2B). Moreover, western blotting showed that sEVs’ markers (CD63 and TSG101) were abundantly expressed, and the main contaminant of serum (Albumin) was rarely expressed in sEVs compared to that in the serum (Figure 2C). Though there were no significant differences in the protein concentrations of sEVs between HC, Cys, and Ca groups, the concentration of the proteins expressed in the Ca group showed a slight increase compared to that of the other two groups (Figure 2D). These results demonstrated that we isolated and purified sEVs from clinical serum samples successfully.

### 3.3. Proteomic Profiles of sEVs

In the present study, 1119 proteins were identified in the TMT-based proteomic analysis of sEVs, among which 996 proteins were available for quantification analysis (Appendix A). Compared to those known vesicular proteins in the Exocarta and Vesiclepedia databases, 85.8% (855/996) proteins were found in previously published data, including the canonical sEVs biomarkers CD9, CD81, CD63, TSG101, flotillin, and syntenin (Figure 3A). With WoLF PSORT software, we found that these proteins were mainly located in the extracellular region (41%), cytoplasm (22%), and plasma membrane (12%) (Figure 3B). To identify the DEPs among the three groups of sEVs, we set the fold change of differential expression as >1.2 and *p*-value < 0.05. Compared to the HC group, the sEVs derived from the Ca group had 201 DEPs, comprising 123 upregulated proteins and 78 downregulated proteins (Appendix A). In the Cys group versus HC group, there were 148 DEPs, comprising 99 upregulated proteins and 49 downregulated proteins (Appendix A). Moreover, 90 DEPs comprising 43 upregulated proteins and 47 downregulated proteins were found in sEVs of the Ca group compared to the Cys group (Appendix A). To better meet the need for liquid biopsy and early diagnosis of ovarian cancer, it is important to identify biomarkers that can distinguish patients with ovarian cancer from patients with benign ovarian tumors and healthy controls. Therefore, we intersected the DEPs of the Ca versus HC group and the DEPs of the Ca versus Cys group to obtain 42 potential candidate protein markers, comprising 20 upregulated proteins and 22 downregulated proteins (Table 1 and Table 2 and Figure 3C,D). Altogether, we successfully identified a series of potential biomarkers of ovarian cancer in the discovery cohort.

### 3.4. Functional Enrichment of DEPs

To provide insights into the biological functions mediated by proteins within sEVs in the pathogenesis of ovarian cancer, 42 DEPs identified above were subjected to GO analysis. The enriched biological processes included the response to stress (detoxification and hemostasis), localization, cell adhesion, coagulation, secretion (exocytosis and vesicle-mediated transport), defense response (innate immune response), and lipid transport. The blood microparticle, extracellular space, fibrinogen complex, and secretory vesicle constituted the main cellular components of DEPs. In addition, molecular function annotations included antioxidant activity, metal ion binding, transporter activity, enzyme regulator activity, and cytokine receptor binding were closely involved among the DEPs (Figure 4A). Based on the REACTOME pathway analyses, the DEPs were enriched in 10 significant signaling pathways, including common pathways of fibrin clot formation, MAPK signaling, integrin signaling, platelet degranulation, and complement activation (Figure 4B). A PPI network was constructed using the STRING database to explore the potential interactions among the 42 DEPs (Figure 4C). Overall, through functional enrichment analyses, we found that the DEPs were involved in the secretion and transportation of the vesicle, the cellular signal transduction, coagulation, and complement activation, implying that circulating sEV proteins derived from patients with ovarian cancer probably actively participate in the regulation of cellular homeostasis, immune response, and tumor growth.

### 3.5. Biomarkers Validation by Targeted Proteomics

We used a multistage refinement workflow for biomarker selection based on the stable expression and functional specificity of proteins (Figure 5A). Based on the robust MS signals and confident quantitation data, a final list of 23 proteins was selected for further validation in an additional cohort comprising 20 healthy controls, 15 patients with cystadenoma, and 50 patients with ovarian cancer (Appendix A). We used median-throughput MS to quantify dozens of protein candidates under the PRM acquisition scheme. The PRM-based targeted proteomics measured 44 tryptic peptides containing unique sequences from 23 protein candidates, among which 22 proteins exhibited reliable and quantitative data across the validation cohort (Appendix A). In addition, the expression profiles revealed significant intergroup differences (Figure 5B). Among these protein candidates, we further screened out 14 potential proteins, comprising 11 upregulated proteins (FGA, FGB, FGG, FGL1, C9, OLFM4, ITIH3, MUC16, HBD, DEFA3, and MPO) and 3 downregulated proteins (APOL1, APOA4, and CD5L), according to the consistency of results in the discovery cohort and the validation cohort. The overall principal component analysis (PCA) model based on these 14 protein candidates demonstrated clear separation between the Ca and HC/Cys groups (Figure 5C). These results suggest that there is a protein profile with high specificity in the serum sEVs of patients with ovarian cancer, which could distinguish patients with ovarian cancer from healthy controls and patients with ovarian cystadenoma.

### 3.6. Development of a Protein Panel for Liquid Biopsy of Ovarian Cancer

The 14 proteins selected above were individually subject to ROC analysis to evaluate their performance to distinguish patients with ovarian cancer from patients with cystadenoma or healthy controls. At this step, eight proteins, comprising six upregulated proteins and two downregulated proteins, were found to have potential diagnostic value (AUC > 0.7) for ovarian cancer (Figure 6A and Table 3). They were involved with the coagulation system (FGA, FGB, FGG, and FGL1), lipid metabolism/transport (APOL1 and APOA4), and protein binding (ITIH3 and MUC16). Univariate logistic regression analysis revealed that the selected six upregulated proteins were risk factors, and two downregulated protein biomarkers were protective factors for patients with ovarian cancer (Table 4). Subsequently, to construct the most simplified model without significant loss of AUC, the diagnostic model was constructed based on the multivariate logistic regression analysis using the stepwise (LR) method (variable was entered if *p* < 0.05 and removed if *p* > 0.1). Finally, a diagnostic model comprising three proteins (FGG, MUC16, and APOA4) was developed (Table 5). Based on the multivariable logistic regression analysis, a diagnostic equation was built as follows: LogitP = 2.481 × FGG + 8.970 × MUC16 − 1.709 × APOA4 − 0.184. ROC analysis revealed that the AUC of the diagnostic model was 0.936 (95% CI, 0.888–0.984), with a sensitivity of 92.0% and a specificity of 82.9%, at the cut-off point of 0.40 (Figure 6B and Table 6). This model exhibits excellent diagnostic performance in advanced ovarian cancer or serous ovarian cancer (Appendix A). It also showed advantages in identifying the relatively rare types of ovarian cancer, such as clear cell carcinoma and endometrioid carcinoma, compared to the serum CA125 level (Appendix A). Moreover, the model yielded an AUC of 0.820 (95%CI, 0.699–0.941), with 71.4% sensitivity and 82.9% specificity in patients with early stage (FIGO stage I and II) ovarian cancer (Figure 6C, Table 6 and Appendix A). In addition, western blot analysis of the three selected sEVs proteins showed consistent results with the targeted proteomics, which indicated that our proteomic analysis was reliable (Appendix A). Taken together, the three-protein panel established here had an excellent diagnostic performance to identify ovarian cancer in a population.

### 3.7. Development of a Novel Model for Differential Diagnosis between Ovarian Cancer and Ovarian Cystadenoma

When an ovarian tumor presents some ambiguous characteristics, the differential diagnosis between cystadenoma and ovarian cancer is essential for therapy. In ROC analysis, eight protein candidates (FGB, FGG, FGL1, MUC16, MPO, APOL1, APOA4, and CD5L) with AUC > 0.65 were subjected to univariate logistic regression analysis. Then, we selected six parameters (FGB, FGG, MUC16, APOL1, APOA4, and CD5L) based on the statistical *p*-value and our experience (Appendix A) for the subsequent multivariate logistic regression analysis. We used the stepwise (LR) method as described before to establish a novel model for accurately identifying ovarian cancer in the presence of a definite ovarian mass suspected to be ovarian cancer or ovarian cystadenoma. Finally, a diagnostic model comprising three proteins (MUC16, APOA4, and CD5L) was developed (Figure 7A and Table 7). The diagnostic equation was as follows: LogitP = 9.468 × MUC16 − 1.887 × APOA4 − 3.260 × CD5L + 4.961. ROC analysis revealed that the AUC of the diagnostic model was 0.915 (95% CI, 0.847–0.983), with a sensitivity of 80.0%, specificity of 93.3%, and accuracy of 83.1% (Table 8). The CA125 level in serum (serumCA125) exhibited similar diagnostic performance in terms of AUC, sensitivity, specificity, and accuracy in ROC analyses (Figure 7A and Table 8). Once serumCA125 was incorporated in the multivariate logistic regression analysis, a novel model (equation: LogitP = 0.009 × serumCA125 − 1.253 × APOA4 − 1.592 × CD5L + 2.348) was developed with serumCA125, APOA4, and CD5L using the LR method (Table 7). The new integrative model showed an AUC of 0.945 (95% CI, 0.890–1.000), sensitivity of 88.0%, specificity of 93.3%, and accuracy of 89.2% (Figure 7A and Table 8). Compared with the single index of serumCA125 in ovarian cancer at early stages (FIGO stage I + II), the AUC of this model was only slightly increased (0.867 (95% CI, 0.733–1.000) versus 0.843 (95% CI, 0.701–0.985)), whereas the sensitivity, specificity, and accuracy of this model were significantly improved (Figure 7B and Table 9). Altogether, the model displayed excellent performance for differential diagnosis between ovarian cancer and ovarian cystadenoma and showed better early-stage ovarian cancer (FIGO I + II) recognition than serumCA125 alone.

## 4. Discussion

In the present study, we performed TMT-based LC-MS/MS analysis to obtain the proteomic profiles of serum sEVs for EOC. Potential biomarkers were identified and further validated via a targeted proteomic approach. A panel composed of three proteins (FGG, MUC16, and APOA4) in serum sEVs showed remarkable performance for ovarian cancer screening in populations. Moreover, a novel panel integrated with serum CA125, sEV-CD5L, and sEV-APOA4 showed potential value for the differential diagnosis between ovarian cancer and ovarian cystadenoma, even at their early stages.

Recently, great advances have been made in ovarian cancer screening and prevention, including the use of tailored prevention and screening methods that combine genetic and epidemiological factors to predict the individual risk of ovarian cancer [19,20]. No definitive mortality reduction was reported for screening compared with no screening until now, suggesting that there is a need to find screening strategies better than CA125 that are noninvasive, simple, and specific for detecting ovarian cancer, especially in its early stage. By far, a major focus of ovarian cancer biomarker discovery is tumor-specific biomarkers in plasma. Examination of various information on tumor cells in the blood represents the new diagnostic tool, namely, liquid biopsy, which quickly identifies biological behaviors of cancer cells, including cell death, clonal evolution, and drug resistance [21,22]. Although many potential benefits of circulating tumor cells (CTCs) have been demonstrated in ovarian cancer patients, the major challenge is processing and analyzing CTCs [23]. In the study conducted by Shao, cell-free DNA carried genetic and epigenetic changes that mimic tumor cells and may be used as a tumor-specific biomarker with higher sensitivity and specificity than CA125 [24]. Inconsistent results have been demonstrated for cfmiRNA in the diagnosis of cancers, which may be attributed to difficulties in sample collection, processing, and RNA stability and quantification, limiting the opportunity to use cfmiRNA as a promising cancer biomarker [25]. EVs provide a protective environment for miRNAs against RNase destruction and are, therefore, considered promising targets for liquid biopsies [26]. Moreover, typical characteristics of sEVs such as size, appearance, specific biomarkers, circulation stability, and the wide availability of sEV isolation kits make it a more suitable target for liquid biopsy than others.

In ovarian cancer, protein and miRNA are the main objects of sEV components considered as biomarkers for diagnosis and/or prognosis and are closely correlated to drug resistance, tumor microenvironment, and immune regulation [27]. We established a three-protein model for liquid biopsy for ovarian cancer with minimal loss of diagnostic efficacy, which showed good diagnostic efficacy even in the early-stage disease. In addition to early diagnosis, differential diagnosis of ovarian masses is also an important reference for preoperative therapeutic options [28]. Elevated CA125 level is associated with other conditions besides ovarian cancer, indicating that the differential diagnosis of benign and malignant ovarian tumors is still worth further optimization [29]. For this purpose, we also established a three-protein panel whose performance is comparable to CA125 using tumor-specific sEVs proteomics. Once serumCA125 was included in the protein candidates, a new integrative model with improved sensitivity and accuracy was generated, with APOA4, CD5L, and serumCA125 replacing sEV-MUC16. This may be due to the poor encapsulation of MUC16 as a macromolecule in sEVs and the limitations of detection methods. Importantly, this new integrated model still has superior AUC, sensitivity, specificity, and accuracy in the differential diagnosis between early-stage ovarian cancer and cystadenoma.

As mediators of cellular communication, sEVs may play the same role in similar biological behaviors of different types of cancer. For example, ALCAM/CD166 mediates the docking and uptake of cancer cell derived EVs and promotes the peritoneal metastasis cascade in colorectal cancer and ovarian cancer [30]. For diagnosis of cancer, biomarkers often contain the characteristic information of different cancer types. A recent study trying to diagnose multiple types of early-stage cancer isolated and purified EVs in plasma to determine the levels of potential protein markers by immunoassay. By utilizing artificial intelligence, 13 EV protein markers were analyzed to predict the likelihood of malignancy [31]. CA19-9 is commonly used to monitor the therapeutic efficacy and recurrence of pancreatic cancer. The novel diagnostic method contains CA19-9 and has excellent diagnostic performance for early-stage pancreatic cancer (95.7% detection rate), while detection of early-stage ovarian cancer and bladder cancer were 74.4% and 43.8% [31], suggesting the importance of characteristic biomarkers for diagnosing specific cancer type. The novel diagnostic protein panel in our study included CA125, a specific marker for ovarian cancer, and its performance in diagnosing ovarian cancer including their early-stage disease and discriminating between benign and malignant ovarian tumors in the population was more effective than CA125. Aiming at identification of sEV proteins as diagnostic biomarkers for ovarian cancer, early studies have shown that some proteins from EVs such as claudin-4, TGF-β1 and MAGE3/6 may have certain diagnostic value in ovarian cancer [15,16], but no further demonstration was made in a larger population sample. A recent study utilized mass spectrometry to detect the specific protein profiles in serum exosomes of ovarian cancer and revealed that the coagulation pathway was significantly enriched and that those cascade-related proteins present diagnostic and prognostic values [32]. Our study also discovers relevant changes in the coagulation pathway, and our conclusions are more convincing because of the larger sample size, the stringent controls, an integrative multi-index model to optimize diagnostic performance, and the validation using targeted proteomics.

Circulating proteins within sEVs carry tumor-specific signals and communicate extensively with tissues throughout the body, thus mediating important events in tumor growth, cancer-associated microenvironment, and tumor evolution [33]. The elucidation of the signaling network can provide new perspectives for the treatment of ovarian cancer. Among the biomarker candidates we screened, MUC16 has been proposed to exert roles in the innate defense of tracheal epithelium or the immune microenvironment of cancer [34]. The FGG gene encodes the gamma chain of fibrinogen, a major component of blood clots, which promotes coagulation. Furthermore, fibrinogen has also been related to leukocyte migration, phagocytosis of microorganisms, tumor growth and metastasis, chemoresistance, and epithelial-mesenchymal transition [35,36,37,38]. APOA4 is not only a lipid-binding protein, but also has a role in the immune response [39,40]. Our model not only reflects the key role of CA125 for diagnosing ovarian cancer, but also suggests that the occurrence and development of ovarian cancer may be accompanied by the remodeling of cellular lipid metabolism and the complex interaction between the tumor and their hosts’ immune system [41,42,43,44]. These findings suggest that these proteins within sEVs found in our study can serve as diagnostic markers for ovarian cancer and provide new perspectives for therapeutic research on ovarian cancer. However, the specific molecular mechanism and the functional roles of sEVs still need to be further studied, complementing the limitations of the current study. The sample size of the study was limited, and more patients should be included to reduce bias and prove the universality of the model. Moreover, it is our next research plan to establish a standardized process for detection of multiple indicators through ELISA and perform integrative analyses of quantitative data. Until now, more research is still needed to collect sufficient serum samples and establish a more robust testing process.

## 5. Conclusions

In summary, we have developed a novel diagnostic panel for ovarian cancer screening and differential diagnosis between benign and malignant ovarian masses based on the proteomic profiles study of serum sEVs. The model has shown robust performance in ovarian cancer, even at the early stage, supplementing the current clinical diagnostic measures and providing a new idea and strategy for diagnosing ovarian cancer.

## Figures and Tables

**Figure 1 cancers-14-03719-f001:**
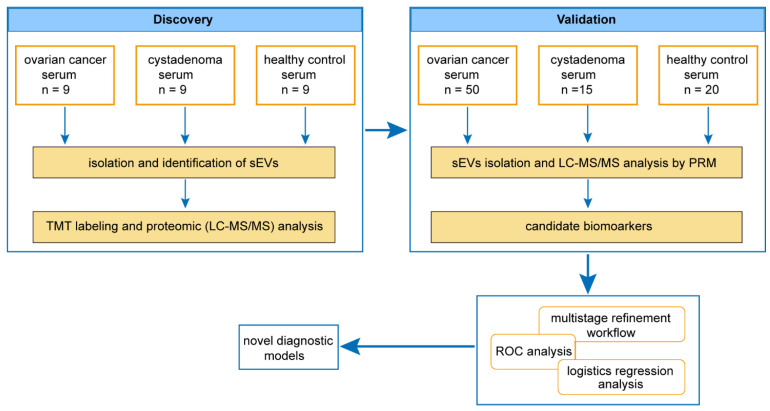
The overall design and flow chart to obtain protein profiles of serum sEVs for ovarian cancer and establish novel diagnostic models. In the discovery study, nine patients with ovarian cancer or cystadenoma and nine healthy controls were enrolled. Isolated sEVs were subjected to TMT labeling LC-MS/MS analysis. PRM was used to further assess candidate biomarkers in a validation cohort composed of 50 patients with ovarian cancer, 15 patients with cystadenoma, and 20 healthy controls. Through ROC analyses, logistic regression analyses and a multistage refinement, novel diagnostic models were constructed for the purpose of ovarian cancer screening in a population or differential diagnosis between ovarian cancer and cystadenoma.

**Figure 2 cancers-14-03719-f002:**
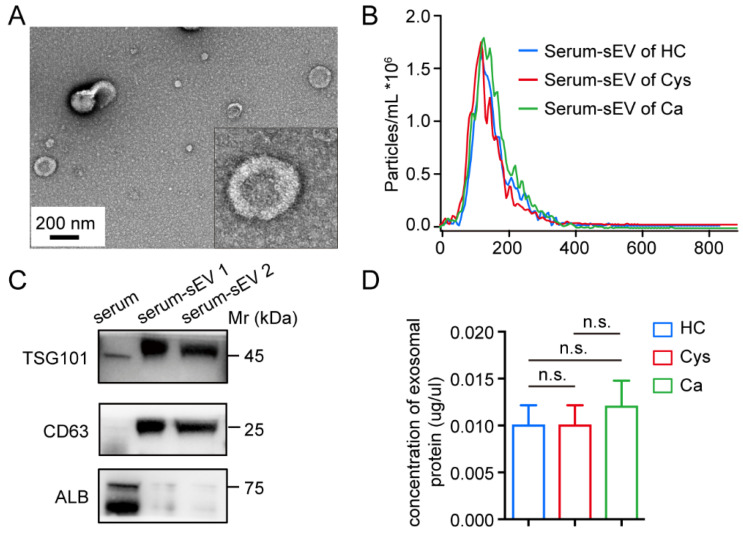
Identification of sEVs. (**A**) The typical morphology of sEVs using TEM; the picture in the lower right corner is the enlarged picture. Scale bar, 200 nm. (**B**) NTA analysis of sEVs from different groups; the horizontal axis represents the diameter of the particle, and the vertical axis represents the number of qualified particles. (**C**) Protein expression of canonical biomarkers of sEVs using western blotting. (**D**) Comparison of the protein concentrations after extraction from different groups of sEVs. n.s., no significance.

**Figure 3 cancers-14-03719-f003:**
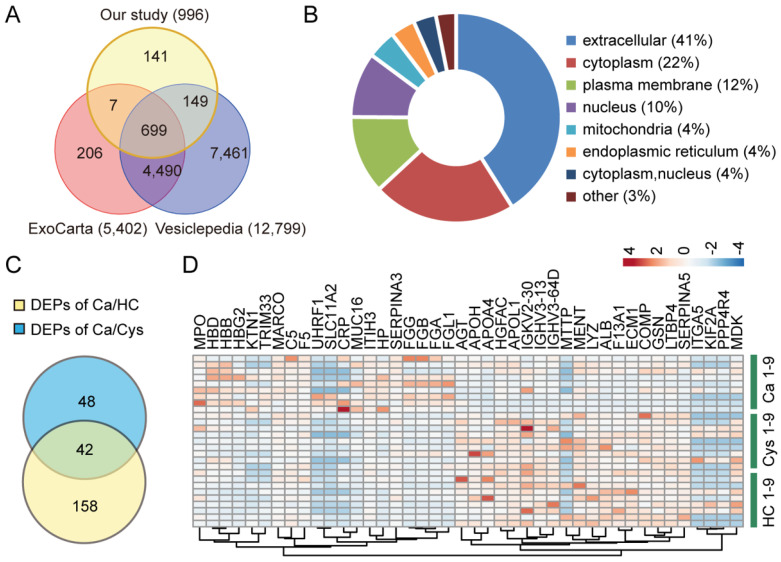
Proteomic profiles of sEVs. (**A**) Venn diagram showed the intersection between the proteins we identified and the proteins from the known databases ExoCarta and Vesiclepedia. (**B**) The circle chart displayed the cellular location of the identified proteins. (**C**) Venn diagram showed the intersection of the DEPs between the Ca group and the HC group and the DEPs between the Ca group and the Cys group. (**D**) Heat map of protein expression distribution of the above 42 DEPs from (**C**).

**Figure 4 cancers-14-03719-f004:**
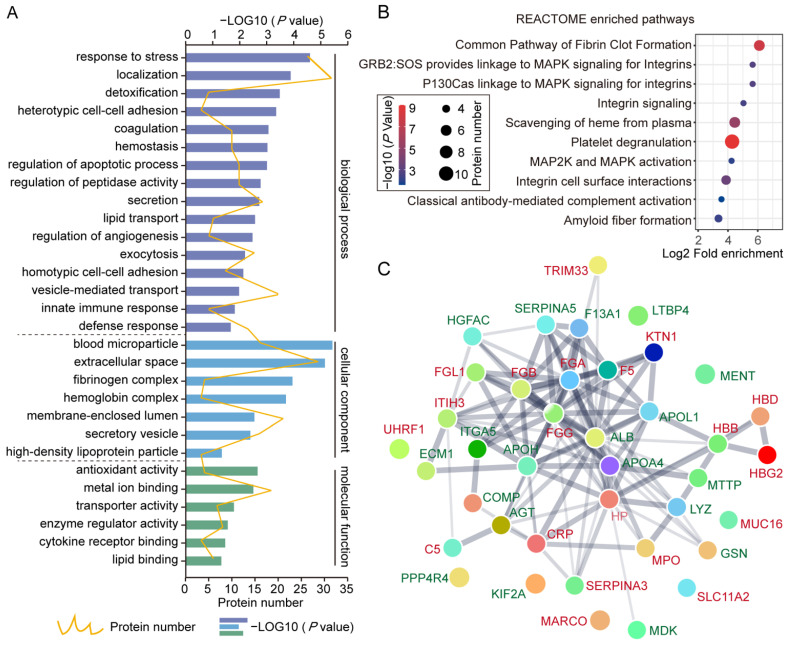
Functional enrichment of DEPs. (**A**) Functional classification chart of DEPs under three GO categories (biological process, cellular component, and molecular function). The yellow wavy line represented the number of proteins enriched to each function, and the bar chart represents the values of statistical *p*-values enriched by each function after LOG10 conversion. (**B**) The above 42 DEPs were included in KEGG pathways analysis; the horizontal axis represents the values of fold change by each pathway after LOG2 conversion, the colored scale bar represents the values of statistical *p*-values enriched by each function after LOG10 conversion, and the black circles represent the number of proteins enriched to each pathway. (**C**) The protein interaction network of DEPs. Protein-to-protein connections were shown in solid lines, and line thickness indicated the strength of data support. The medium confidence was 0.40.

**Figure 5 cancers-14-03719-f005:**
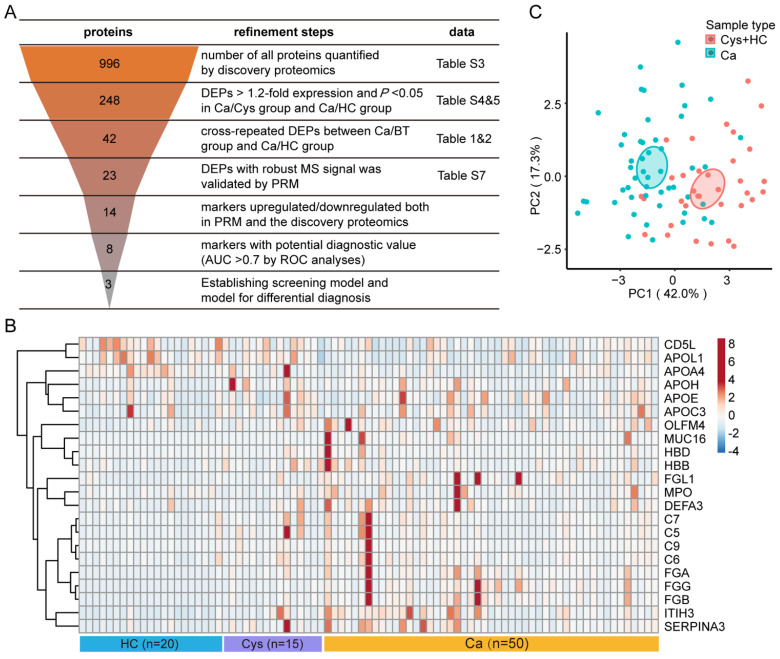
Biomarkers validation by targeted proteomics. (**A**) The funnel diagram illustrates the main processes of narrowing the potential candidates and, finally, establishing a diagnostic model. The orange funnel-shaped picture in the left column represents the process of step-by-step screening of protein candidates, and the numbers inside represent the number of proteins in each step. The middle column represents the performed screening conditions, and the right column represents the corresponding data tables. (**B**) Heatmap of relative abundance of 14 candidate proteins using PRM approach. Colored red represents an increasing trend, colored blue represents a decreasing trend, and the right scale bar represents the fold change of the relative quantitation data. (**C**) PCA plot based on the quantification of 14 proteins based on PRM. The multi-dimensional variables are reduced by the statistical method of PCA and converted into two-dimensional variables with high precision.

**Figure 6 cancers-14-03719-f006:**
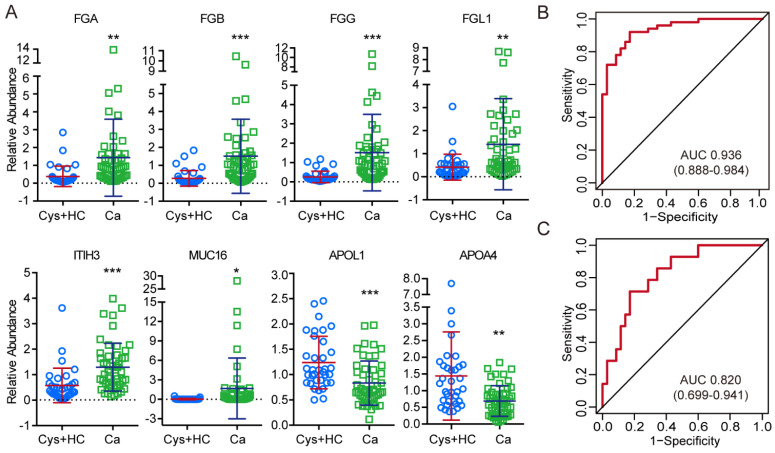
Development of a protein panel for liquid biopsy of ovarian cancer. (**A**) Scatter plots of quantitative comparisons between the Ca group and the HC/Cys group for indicators with potential diagnostic value for ovarian cancer. The blue circles represent relative quantification values for cystadenomas and healthy controls. Green boxes represent relative quantification values for ovarian cancer patients. * *p* < 0.05, ** *p* < 0.01, *** *p* < 0.001. (**B**) ROC analysis of the novel model to diagnose ovarian cancer in a general population. Multivariate logistic regression analysis was performed in all patients to establish a screening diagnostic model followed by receiver operating curve analysis. (**C**) ROC analysis of the novel diagnostic model to diagnose early-stage ovarian cancer.

**Figure 7 cancers-14-03719-f007:**
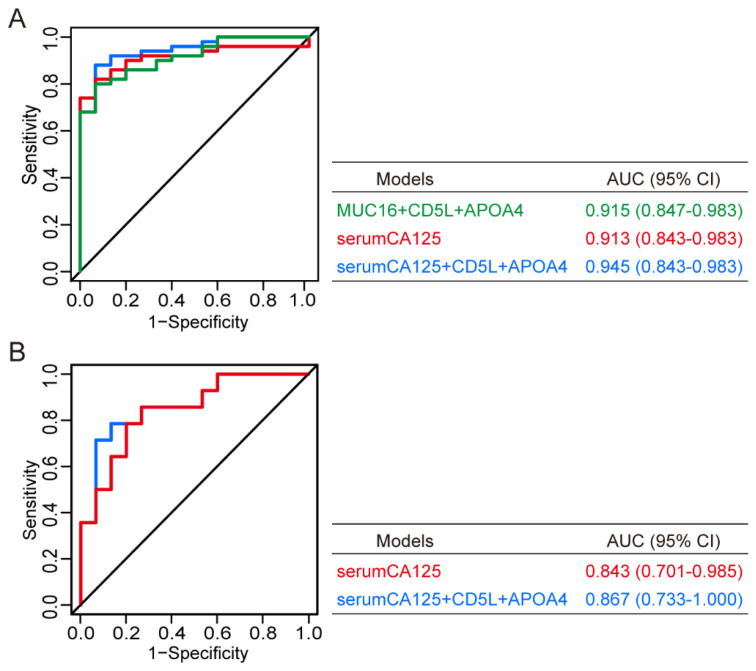
Development of a novel model for differential diagnosis between ovarian cancer and ovarian cystadenoma. (**A**) ROC analyses of serumCA125, a panel comprising MUC16, CD5L, and APOA4 in sEVs, and a panel comprising serumCA125, CD5L, and APOA4 in sEVs. The protein biomarkers in sEVs and serum CA125, or their combination to establish a diagnostic model, analyze their performance in the differential diagnosis of cystadenoma and ovarian cancer. (**B**) Comparison of serumCA125 and the novel panel comprising serumCA125, CD5L, and APOA4 in sEVs for patients with early-stage ovarian cancer.

**Table 1 cancers-14-03719-t001:** The upregulated DEPs existed in both Ca/Cys group and Ca/HC group (>1.2 fold).

No.	Protein Accession	Protein Description	Gene Name	Ca/(Cys + HC) Ratio
1	Q96T88	Ubiquitin-like with PHD and ringer finger domains 1	UHRF1	3.79
2	P02042	Hemoglobin subunit delta	HBD	1.81
3	Q86UP2	Kinectin 1	KTN1	2.30
4	Q9UEW3	Macrophage receptor with collagenous structure	MARCO	1.22
5	Q06033	Inter-alpha-trypsin inhibitor heavy chain 3	ITIH3	1.60
6	P05164	Myeloperoxidase	MPO	1.71
7	P49281	Solute carrier family 11 member 2	SLC11A2	5.81
8	P02679	Fibrinogen gamma chain	FGG	2.73
9	P01031	Complement C5	C5	1.39
10	P02675	Fibrinogen beta chain	FGB	2.30
11	P02741	C-reactive protein	CRP	6.00
12	P01011	Serpin family A member 3	SERPINA3	1.33
13	P12259	Coagulation factor V	F5	1.27
14	Q8WXI7	Mucin-16	MUC16 *	2.69
15	P69892	Hemoglobin subunit gamma-2	HBG2	2.02
16	P02671	Fibrinogen alpha chain	FGA	2.14
17	Q9UPN9	Tripartite motif containing 33	TRIM33	1.80
18	P68871	Hemoglobin subunit beta	HBB	1.96
19	P00738	Haptoglobin	HP	2.63
20	Q08830	Fibrinogen-like protein 1	FGL1	2.45

* Also known as CA125. Generally, CA125 is determined by ELISA kits, and the CA125 levels in this study are examined by immunoassay in clinical practice. In our study, CA125 levels were measured by mass spectrometry. We defined CA125 in EV as MUC16 to differentiate it from CA125 in serum.

**Table 2 cancers-14-03719-t002:** The downregulated DEPs existed in both Ca/Cys group and Ca/HC group (<1/1.2 fold).

No.	Protein Accession	Protein Description	Gene Name	OVCa/BT + HC Ratio
1	P01019	Angiotensinogen	AGT	0.67
2	P61626	Lysozyme C	LYZ	0.75
3	P55157	Microsomal triglyceride transfer protein	MTTP	0.41
4	Q04756	Hepatocyte growth factor activator	HGFAC	0.79
5	P08648	Integrin alpha-5	ITGA5	0.71
6	P02749	Apolipoprotein H	APOH	0.69
7	P06310	Immunoglobulin kappa variable 2-30	IGKV2-30	0.50
8	O00139	Kinesin family member 2A	KIF2A	0.59
9	Q6NUP7	Protein phosphatase 4 regulatory subunit 4	PPP4R4	0.51
10	O14791	Apolipoprotein L1	APOL1	0.65
11	P49747	Cartilage oligomeric matrix protein	COMP	0.78
12	P00488	Coagulation factor XIII A chain	F13A1	0.63
13	P06396	Gelsolin	GSN	0.80
14	P01766	Immunoglobulin heavy variable 3-13	IGHV3-13	0.70
15	A0A0J9YX35	Immunoglobulin heavy variable 3-64D	IGHV3-64D	0.59
16	Q9BUN1	Chromosome 1 open reading frame 56	MENT	0.69
17	P06727	Apolipoprotein A-IV	APOA4	0.47
18	Q16610	Extracellular matrix protein 1	ECM1	0.63
19	P02768	Albumin	ALB	0.61
20	P21741	Midkine	MDK	0.60
21	Q8N2S1	Latent-transforming growth factor beta-binding protein 4	LTBP4	0.79
22	P05154	Serpin family A member 5	SERPINA5	0.62

**Table 3 cancers-14-03719-t003:** Area under the ROC curve of proteins for diagnosis of ovarian cancer.

Proteins	AUC	*p*-Value	95% Confidence Interval (CI)	
			Lower Bound	Upper Bound
FGA	0.814	<0.001	0.717	0.910
FGB	0.856	<0.001	0.774	0.938
FGG	0.879	<0.001	0.808	0.951
FGL1	0.741	<0.001	0.635	0.846
C9	0.698	0.002	0.582	0.815
OLFM4	0.638	0.031	0.519	0.757
ITIH3	0.783	<0.001	0.683	0.883
MUC16	0.807	<0.001	0.716	0.898
HBD	0.629	0.045	0.507	0.750
DEFA3	0.670	0.008	0.553	0.787
MPO	0.579	0.218	0.457	0.700
APOL1	0.744	<0.001	0.640	0.848
APOA4	0.746	<0.001	0.643	0.850
CD5L	0.673	0.007	0.558	0.788

**Table 4 cancers-14-03719-t004:** Univariate logistic regression analysis for the prediction of ovarian cancer.

	Univariate Analysis	
Parameters	OR (95%CI)	*p*-Value
FGA	5.04 (1.80–14.09)	0.002
FGB	9.10 (2.79–29.74)	<0.001
FGG	22.48 (4.61–109.51)	<0.001
FGL1	3.26 (1.37–7.74)	0.007
ITIH3	4.03 (1.72–9.44)	0.001
MUC16	595.52 (4.53–9038.36)	0.01
APOL1	0.170 (0.06–0.48)	0.001
APOA4	0.195 (0.08–0.49)	0.001

**Table 5 cancers-14-03719-t005:** The 3-proteins logistic regression model for ovarian cancer diagnosis.

Proteins	Functions	Multivariate Analysis	
		Coef.	*p*-Value
FGG	Blood coagulation cascade, platelet activation	2.481	0.005
MUC16	Cell adhesion, up-regulated in ovarian cancer cells	8.970	0.037
APOA4	Chylomicrons and VLDL secretion and catabolism Lipid transporter activity	−1.709	0.014
constant		−0.184	0.801

**Table 6 cancers-14-03719-t006:** Area under the ROC curve of the 3-proteins diagnostic panel in validation cohort.

		Validation Cohort			Validation Cohort with Early Stage		
	Cut-Off	AUC (95%CI)	SN (%)	SP (%)	AUC (95%CI)	SN (%)	SP (%)
3-proteins model	0.40	0.936 (0.888–0.984)	92.0	82.9	0.820 (0.699–0.941)	71.4	82.9

SN, sensitivity; SP, specificity.

**Table 7 cancers-14-03719-t007:** The logistic regression model for differential diagnosis of ovarian masses.

Proteins	Multivariate Analysis		Variables	Multivariate Analysis	
	Coef.	*p*-Value		Coef.	*p*-Value
MUC16	9.468	0.041	serumCA125	0.009	0.060
CD5L	−3.260	0.005	CD5L	−1.592	0.122
APOA4	−1.887	0.030	APOA4	−1.253	0.175
constant	4.961	0.002	constant	2.348	0.153

**Table 8 cancers-14-03719-t008:** The evaluation indicators of the differential models in patients with ovarian masses.

	Cut-Off	SN (%)	SP (%)	Accuracy (%)
MUC16 + CD5L + APOA4	0.79	80.0	93.3	83.1
serumCA125	35	84.0	86.7	84.6
serumCA125 + CD5L + APOA4	0.58	88.0	93.3	89.2

**Table 9 cancers-14-03719-t009:** The evaluation indicators of the model for diagnosis of ovarian cancer at early stages.

	Cut-Off	SN (%)	SP (%)	Accuracy (%)
serumCA125	35	57.1	86.7	72.4
serumCA125 + CD5L + APOA4	0.58	71.4	93.3	82.8

## Data Availability

The data presented in this study are available on request from the corresponding author.

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
