# Peer review of "Protein Panel of Serum-Derived Small Extracellular Vesicles for the Screening and Diagnosis of Epithelial Ovarian Cancer"

_cancers, 2022, doi:10.3390/cancers14153719_

Round 1

Reviewer 1 Report

Lai et al analyzed the circulating small extracellular vesicles (sEVs) derived from the serum of healthy humans, patients with cystadenoma, and ovarian cancer through an unbiased TMT-LC-MS/MS approach, and identified multiple differentially expressed genes as potential non-invasive diagnostic markers for ovarian cancer. This study is potentially clinically significant and of interest to the research community on ovarian cancer.  However, there are some major concerns that the authors should address prior to publication.

Major points:

1) There are 50 patients with ovarian cancer enrolled in the current study. Information about these patients, such as disease stage and age, should be included in the methods section, considering patient information is valuable when identifying diagnostic markers.

2) Line 261-262, the authors set the threshold for the differential expression fold change as >1.2. I am concerned with this, as the changes investigated here are quite marginal. 

3) Validation of key proteins by ELISA or western blots in Table 4 (or at least in Table 5). 

4) How about the clinical relevance/significance of FGG, MUC16, APOA4, and CD5L in ovarian cancer from some public databases (TCGA, etc)?

5) Please include some discussion upon the potential uniqueness of these 3 biomarkers for ovarian cancer but not for other types of cancer.

Minor points:

1) format some citations (line 449, 463, etc). It should be read as CA125ref22.

Author Response

Dear Reviewer,

We are grateful for your effort towards our manuscript. Those comments are of great help to revise and improve our paper. We would like to make the following responses based on the reviewer’ comments.

Reviewer1

Major points:

1) There are 50 patients with ovarian cancer enrolled in the current study. Information about these patients, such as disease stage and age, should be included in the methods section, considering patient information is valuable when identifying diagnostic markers.

Response: We have provided information including age, disease stage, CA125 levels, grade, and histological type in supplementary Table 1 (Line 108-109). We also made full use of valuable clinical information in the study, such as disease stage and CA125 levels. We analyzed the diagnostic performance of the novel model in different clinical stages and pathological types of ovarian cancer (Figure S2). At the same time, the diagnostic value of combination of serum CA125 and sEV proteins was analyzed (Line 408-416).

2) Line 261-262, the authors set the threshold for the differential expression fold change as >1.2. I am concerned with this, as the changes investigated here are quite marginal.

Response: There is currently no hard and fast rule for the fold change of differentially expressed proteins in proteomics. Most of the published literatures were analyzed according to the fold changes of 1.2-2.0 times. Generally reliable biomarkers need to be validated by other methods after being screened by the fold change from proteomic data. For example, researchers used mass spectrometry to compare the cytosolic proteomes from the wild-type and AAC2A128P cells and concluded that AAC2A128P expression downregulated 107 proteins by >2-fold and upregulated 297 proteins by >1.266-fold in an article titled “A cytosolic network suppressing mitochondria-mediated proteostatic stress and cell death” (Nature. 2015 Aug 27; 524 (7566), 481-4).

In our research, we set the fold difference at 1.2-fold in the label-free proteomics in order not to miss important potential biomarkers in the screening stage due to the variability between clinical samples. Subsequently, we verified those proteins obtained from the preliminary screening by targeted proteomics to obtain stable and reliable protein biomarkers. 

3) Validation of key proteins by ELISA or western blots in Table 4 (or at least in Table 5).

Response: According to the reviewer’s suggestion, we utilized western blots for the validation of key proteins included in our models and the results are present in supplementary figure 3 (Page 12, Line 375-377).

4) How about the clinical relevance/significance of FGG, MUC16, APOA4, and CD5L in ovarian cancer from some public databases (TCGA, etc)?

Response: This research aims to identify specific protein biomarkers in the extracellular vesicles (EVs) obtained from the serum of patients with ovarian cancer. Most databases such as TCGA focus on levels of RNA or protein expressed in tissues, and therefore, differentially expressing proteins including FGG, MUC16, APOA4 and CD5L cannot be directly validated in these databases. The databases for proteins from EVs, including ExoCarta, Vesiclepedia, EVpedia, etc., provide abundant proteomic data for different diseases, but they generally lack rigorous comparison group, or clinicopathological features such as prognosis and recurrence. Therefore, it is temporarily impossible to analyze the clinical relevance of FGG, MUC16, APOA4 and CD5L in EVs directly in the public databases, which may be possible with the gradual in the future with the gradual improvement of the database.

5) Please include some discussion upon the potential uniqueness of these 3 biomarkers for ovarian cancer but not for other types of cancer.

Response: With this suggestion of the reviewer, we have modified the discussion part, and believed that the revised discussion can displayed the content, significance, and merits of this study more clearly and profoundly (Page 18, Line 509-531).

Minor points:

1) format some citations (line 449, 463, etc). It should be read as CA125ref22.

Response: We have updated the format of the references as requested (Line 458).

Reviewer 2 Report

This manuscript by Lai and coworkers describes the proteomic identification of proteins contained in small extracellular vesicles (sEVs) obtained from women with ovarian cancer, benign gynecological disease, or control subjects.  The study evaluated 50 patients and 29 healthy subjects.  The study is carefully performed; however the manuscript would be improved by the following:

1.       Line 217 – what is the rationale for including patients from the discovery cohort in the validation cohort?  This is an unusual way to evaluate the data. How do the results change if the discovery cohort patients are removed from the validation cohort?

2.      In general, the manuscript would be improved by inclusion of a discussion as to how the proteins identified in sEVs from ovarian cancer patients in this study compare to other proteomic analyses of ovarian cancer sEVs.  Are similar or unique proteins found here?

3.      The authors describe a good sensitivity and specificity using a 3-protein diagnostic model when comparing ovarian cancer to controls.  However it is not discussed whether the test is specific to ovarian cancer relative to other cancers that may be prevalent in the peritoneal cavity.  Inclusion of patients with pancreatic or colorectal cancer would be informative, as would inclusion of patients with peritonitis.

4.      As mentioned, diagnosis of early stage disease is a key clinical problem in ovarian cancer.  The manuscript would be improved if the authors would clarify in the main manuscript how many stage I/II patients were included in the analyses. 

5.      Line 398, the authors state that they selected 6 parameters based on the statistical p value “and our experience” – what is meant by the latter part of the statement?

6.      The manuscript should clarify the relationship between MUC16 and CA125.

Author Response

Dear Reviewer,

We are grateful for your effort towards our manuscript. Those comments are of great help to revise and improve our paper. We would like to make the following responses based on the reviewer’ comments.

Reviewer2

1) Line 217 – what is the rationale for including patients from the discovery cohort in the validation cohort? This is an unusual way to evaluate the data. How do the results change if the discovery cohort patients are removed from the validation cohort?

Response: First, EVs samples isolated from the serum of patients with ovarian cancer were difficult to obtain. Then the protein expression trend of the discovery cohort was verified again by the targeted proteomics in the validation cohort. The method of validation was not the same as that of the discovery cohort, which had high specificity and can better explain the real phenomenon. Moreover, the proportion of samples from the discovery cohort to the validation cohort was small. Taking into consideration all of the above reasons, we included patients from the discovery cohort in the validation cohort and convinced that the conclusions were valid and reliable.

2) In general, the manuscript would be improved by inclusion of a discussion as to how the proteins identified in sEVs from ovarian cancer patients in this study compare to other proteomic analyses of ovarian cancer sEVs. Are similar or unique proteins found here?

Response: As suggested by the reviewer, we have revised the discussion. We discussed and compared our study with early studies on sEVs proteomics in ovarian cancer (Page 17-18, Line 495-508).

3) The authors describe a good sensitivity and specificity using a 3-protein diagnostic model when comparing ovarian cancer to controls. However, it is not discussed whether the test is specific to ovarian cancer relative to other cancers that may be prevalent in the peritoneal cavity.  Inclusion of patients with pancreatic or colorectal cancer would be informative, as would inclusion of patients with peritonitis.

Response: Thanks for your suggestion. As mediators of cellular communication, sEVs may play the same role in similar biological behaviors of different types of cancer. For diagnosis of cancer, biomarkers often contain the characteristic information of different cancer types. However, there were also studies that suggested the importance of characteristic biomarkers for diagnosing specific cancer type. We elaborated on this point in the discussion section (Page 17, Line 482-494).

4) As mentioned, diagnosis of early stage disease is a key clinical problem in ovarian cancer.  The manuscript would be improved if the authors would clarify in the main manuscript how many stage I/II patients were included in the analyses. 

Response: The detailed information of all the patients is provided in Table S1, which included the clinical stages. We also added this in the article. Of the 50 ovarian cancer patients, the pathological stages were distributed as follows: stage I in 12 patients, stage II in 2 patients, stage III in 25 patients, and stage IV in 11 patients (Page 5, Line 214-216).

5) Line 398, the authors state that they selected 6 parameters based on the statistical p value “and our experience” – what is meant by the latter part of the statement?

Response: In the study, when we established the novel model for differential diagnosis between ovarian cancer and ovarian cystadenoma, the sample size gets smaller. Eight protein candidates (FGB, FGG, FGL1, MUC16, MPO, APOL1, APOA4, and CD5L) were subjected to univariate logistic regression analysis. We set the p-value < 0.1 in univariate logistic regression analysis to enter the subsequent multivariate logistic regression analysis. The p-value for MUC16 is 0.109, and MUC16 is currently recognized as the most widely used ovarian cancer-specific biomarker. In order to reduce the loss of effective information, we still included it in the next multivariate logistic regression analysis based on experience. So, we stated that “we selected six parameters based on the statistical p value and our experience” in the manuscript.

6) The manuscript should clarify the relationship between MUC16 and CA125.

Response: We apologize for not clarifying this point clearly in the article. MUC16 is also known as CA125. Generally, CA125 is determined by ELISA kits, and the CA125 levels in this study are examined by immunoassay in clinical practice. In our study, CA125 levels were measured by mass spectrometry. We defined CA125 in EV as MUC16 to differentiate it from CA125 in serum. To avoid ambiguity, we explained this point in the manuscript (Page 7, Line 276-278).

Round 2

Reviewer 1 Report

The manuscript has been improved substantially. It is ready for publication in Cancers.